# Spatiotemporal Analysis and Risk Assessment Model Research of Diabetes among People over 45 Years Old in China

**DOI:** 10.3390/ijerph19169861

**Published:** 2022-08-10

**Authors:** Zhenyi Wang, Wen Dong, Kun Yang

**Affiliations:** 1Faculty of Geography, Yunnan Normal University, Kunming 650500, China; 2GIS Technology Engineering Research Centre of West-China Resources and Environment of Educational Ministry, Yunnan Normal University, Kunming 650500, China

**Keywords:** spatiotemporal analysis, risk factors, binary logistic regression, random forest model

## Abstract

Diabetes, which is a chronic disease with a high prevalence in people over 45 years old in China, is a public health issue of global concern. In order to explore the spatiotemporal patterns of diabetes among people over 45 years old in China, to find out diabetes risk factors, and to assess its risk, we used spatial autocorrelation, spatiotemporal cluster analysis, binary logistic regression, and a random forest model in this study. The results of the spatial autocorrelation analysis and the spatiotemporal clustering analysis showed that diabetes patients are mainly clustered near the Beijing–Tianjin–Hebei region, and that the prevalence of diabetes clusters is waning. Age, hypertension, dyslipidemia, and smoking history were all diabetes risk factors (*p* < 0.05), but the spatial heterogeneity of these factors was weak. Compared with the binary logistic regression model, the random forest model showed better accuracy in assessing diabetes risk. According to the assessment risk map generated by the random forest model, the northeast region and the Beijing–Tianjin–Hebei region are high-risk areas for diabetes.

## 1. Introduction

### 1.1. Background

In the past few decades, diabetes has become one of the most common chronic noncommunicable diseases in both developed and developing countries [1]. Diabetes is emerging as an epidemic all over the world, and it is a common chronic disease that seriously threatens human health [2]. It affects the quality of lives of many people around the world [3], and the quality of life for Chinese residents is also affected by diabetes. China has a large and rapidly growing elderly population. Studies have shown that diabetes may also lead to the occurrence of other diseases, such as metabolic-associated fatty liver disease [4,5,6,7,8]. Diabetes has become another serious health hazard, following cardiovascular and cerebrovascular diseases and tumors. Half (50.1%) of the population does not even know if they are diabetic, which greatly increases the global disease burden [9]. According to data published by the International Diabetes Federation (IDF), the prevalence of diabetes is increasing rapidly around the world. According to IDF estimates, the prevalence of diabetes in China has reached 10.6%, with the proportion of undiagnosed diabetics as high as 51.7% [10].

Disease mapping has been historically considered one of the most important public health issues, derived from an understanding of the relationship between health and location. Understanding this relationship has been the goal of scientists and researchers for decades [11]. Geographic information systems (GIS) are a type of computer software used for data capturing, thematic mapping, updating, retrieving, structured querying, and analyzing the distribution and differentiation of various phenomena, including communicable and non-communicable diseases across the world, with reference to various periods [12]. The most important characteristic of a geographic information system is its powerful spatial analysis function. Nowadays, geographic information systems have played an irreplaceable role in many aspects of daily life. A GIS is, at its heart, a simple extension of statistical analyses that joins epidemiological, sociological, clinical, and economic data with references to space [13].

### 1.2. Research Status

The GIS approach has the potential for broader applications within public health program evaluation [14,15]. With the rapid development of GIS systems and related technologies, the advantages that GIS provides for the study of chronic diseases have been gradually recognized, and the application scope has also transitioned from infectious diseases to chronic diseases [16,17]. Some scholars have applied GIS to diabetes research and proposed that geospatial methods should be a part of diabetes research because many pathogenic pathways have inherent spatial properties [18]. GIS can be used to map the geographical distribution of disease prevalence, the trend of disease transmission, and the spatial modeling of environmental factors influencing disease occurrence [11].

Although diabetes is a health threat all over the world, its prevalence and trends in various countries and regions are heterogeneous [19]. Previous studies have showed that the prevalence of diabetes among middle-aged and elderly people in the central and eastern regions is higher than in the western regions, but the gap was closing [20]. At present, studies on diabetes in Chinese people over 45 years old are mostly regional or related to a single province, but the number of nationwide studies is lacking [18]. Moreover, GIS is seldom used to study the spatial patterns of diabetes [21]. Recent studies in the health field have adopted machine learning and deep learning algorithms. Since machine-learning approaches perform well in predicting diabetes, they are gaining traction in the health profession [22,23]. This research hoped to analyze the regional differences of diabetes among people over 45 years old in China, and to assess diabetes risk [24], thereby aiming to provide reference for the formulation of diabetes prevention and treatment programs.

## 2. Materials and Methods

### 2.1. Data Source

This study is based on the baseline data of the China Health and Retirement Longitudinal Study (CHARLS). The China Health and Retirement Longitudinal Study is part of a worldwide pension tracking survey. This database is one of the most commonly used databases in China to study the health of the middle-aged and older population, and provides high-quality microdata representing households and individuals aged ≥45 years in China. Many scholars have obtained many reliable research results based on CHARLS [25,26,27,28].

The China Health and Retirement Longitudinal Study (CHARLS) aims to collect a high-quality nationally representative sample of Chinese residents ages 45 and older to serve the needs of scientific research on the elderly. The baseline national wave of CHARLS was established in 2011 and includes about 10,000 households in 125 prefecture-level city and 450 villages/resident committees. CHARLS adopts multi-stage stratified probability-proportional-to-size sampling. CHARLS is based on the Health and Retirement Study (HRS) and on related aging surveys such as the English Longitudinal Study of Aging (ELSA) and the Survey of Health, Aging and Retirement in Europe (SHARE) [29].

### 2.2. Diabetes Definition

Prevalence refers to the proportion of the total number of people who have the disease at a specific point in time in a given place. Diabetes was defined as: fasting glucose level ≥ 126 mg/dL (7.0 mmol/L), or 2-h glucose level ≥ 200 mg/dL (11.1 mmol/L), or on medications for high blood sugar, or self-reported diagnosis of diabetes by a physician.

### 2.3. Methods

#### 2.3.1. Spatial Autocorrelation

Global Spatial Autocorrelation statistics are often expressed as Moran’s I (Equation (1)). According to the literature, the classical Moran’s index of Spatial Autocorrelation has been widely used in many knowledge fields, such as epidemiology, ecology, and economics [30]. The index was used to explore the overall spatial pattern of disease prevalence. When the Moran index is between 0 and 1, it indicates that there is a positive correlation between geographical entities. The larger the value, the more obvious the spatial correlation. When the Moran index is between −1 and 0, there is a negative correlation. The smaller the Moran index, the greater the spatial difference. A value of 0 indicates no correlation. In addition, the value also needs to pass the hypothesis test, without which, the Moran index is meaningless.
(1)$$I=\frac{\sum_{i}\sum_{j}W_{ij}Z_{i}Z_{j}/S_{0}}{\sum_{j}Z_{i}^{2}/n}$$
where *Z_i_* = *y_i_* − *ӯ*, where *ӯ* is the mean of the variable *y* representing the observations under study, *W_ij_* is the spatial weight between feature *i* and *j*, and *S*_0_ is the sum of all the elements in the spatial weights matrix (*S*_0_ = ∑*i*∑*j W_ij_*) [31].

Getis and Ord’s G* assessed localized patterns of spatial association. Specifically, Getis and Ord’s G* can indicate regions where low values are clustered (G* > 0) and regions where high values are clustered (G* < 0) [32]. Local Spatial Autocorrelation can accurately indicate the aggregation mode of each spatial unit [33]. Generally, Local Spatial Autocorrelation analysis (LISA) is used. LISA had five results of “high-high” (H-H), “low-low” (L-L), “low-high” (L-H), “high-low” (H-L), and no statistical significance [34]. Respectively, the regions with high prevalence surround the regions with high prevalence, the regions with low prevalence surround the regions with low prevalence, the regions with low prevalence surround the regions with high prevalence and the regions with high prevalence surround the regions with low prevalence. In this study, Moran’s I and LISA plots were calculated for the prevalence of diabetes in members of the Chinese population over 45 years old in 2011, 2013, 2015, and 2018, respectively. ArcGIS 10.4 software (ESRI Inc., Redlands, CA, USA) was used in this study.

#### 2.3.2. Spatial Cluster Analysis

Temporal, spatial, and spatiotemporal scan statistics are now commonly used for disease cluster detection and assessment for a variety of diseases, including cancer, Creutzfeldt–Jakob disease, granulocytic ehrlichiosis, sclerosis, and diabetes. Spatial clustering analysis was performed using SaTScan software (Martin Kulldorff, Harvard Medical School, Boston and Information Management Services Inc, Calverton, MD, USA) to detect spatially clustered areas or high-risk areas of diabetes in members of the Chinese population over 45 years old. The “purely spatial analysis” and “space time analysis” were used to test whether the prevalence of diabetes was randomly distributed in space. To avoid preselection bias as described in the SaTScan User Guide (version 9.1) [35], a maximum spatial cluster size of 10% of the population at risk was used.

#### 2.3.3. Binary Logistic Regression

Binary logistic regression is a linear regression analysis in which the dependent variable is a binary classification variable, requiring logit transformation of the target probability first, so as to ensure that when the probability is at (0, 1), the logit transformation value can be any real number, avoiding the structural defects of the linear probability model. The probability of each classification of a classification variable can be predicted by logistic regression. The dependent variable is a classification variable, and the independent variable can be an interval variable, a classification variable, or a mixture of the interval and the classification variable. Binary logistic regression model is a regression model established for binary variables, such as Equation (2) [36], which can capably meet the modeling requirements of classified data. It has become a commonly used modeling method for classifying variables and has been widely used in many fields, such as medicine. We used IBM SPSS Statistics 26 software(IBM Corp., Armonk, NY, USA) and the test level α = 0.05 was used in this study.
(2)$$ln\frac{p}{1-p}=\beta_{0}+\beta_{1}X_{1}+\beta_{2}X_{2}+\ldots \beta_{i}X_{i}$$

Suppose a survey of diabetes for conditional probability *P_i_* = *P* (*Y_i_* = 1|*X_i_*), according to the type of binary logistic regression model assumes that the probability expression as shown in Equation (3).
(3)$$P_{i}=\frac{1}{1+e^{\left(\beta_{0}+\beta_{1}X_{1}+\beta_{2}X_{2}+\ldots \beta_{i}X_{i}\right)}}=\frac{1}{1+e^{-\left(\beta_{0}+\sum^{\;}\beta_{i}X_{i}\right)}}$$

#### 2.3.4. Geographically Weighted Regression

The geographically weighted regression (GWR) (Equation (4)) is a statistical technique that is used to model heterogeneous spatial processes. It has high accuracy in analyzing location-affected relationships [37].
(4)$$y_{i}=\beta_{0}\left(u_{i},v_{i}\right)+\overset{n}{\underset{k=1}{\sum }}\beta_{k}\left(u_{i},v_{i}\right)x_{i}k+\varepsilon_{i}$$
where (*u_i_*, *v_i_*) denotes the coordinates of the *i*-th point in space, *β_k_* (*u_i_*, *v_i_*) is the regression coefficient of each variable at point *i*, *β*_0_ (*u_i_*, *v_i_*) is a constant term, *ε_i_* is the random error term at point *i*, and *n* is the number of independent variables.

GWR is a local modeling tool based on the optimization of global regression models, which complements the global model by providing a set of coefficients for each geographic unit to determine the spatial variability of the observations [38]. GWR was used to explore the spatial heterogeneity of risk factors in this study.

#### 2.3.5. Random Forest Model

The random forest algorithm can deal with nonlinear problems, has good anti-noise ability, and tends to avoid overfitting. Compared with the traditional multiple linear regression model, the random forest algorithm does not need to set the function form in advance and overcome the complex interaction between covariables [39]. The building blocks of the decision tree-based modeling approach, the random forest model, are bootstrapped and are called bagged aggregates. Random forest models randomly use bagging to identify features, thereby separating each node by selecting the most critical possible to assess or predict variables, which will improve the model’s accuracy without causing overfitting. At present, the random forest model has been widely applied to predict and assess soil moisture, shallow water level, hydrology, and environmental management. In a random forest, factors with a significant influence on logistic regression are included as independent variables into random forest modeling [40], and the presence of diabetes is set as the dependent variable. The total data are divided into a training set and test set according to 7:3. The model parameters are trained through the training set for the assessment of the test set.

## 3. Results

### 3.1. Statistical Analysis and Spatial Distribution

In 2011, a total of 20,525 samples were included, including 1088 cases, with a prevalence of 5.30%. In 2013, a total of 20,525 samples were included, including 1333 cases, with a prevalence of 6.49%; In 2015, a total of 20,525 samples were included, including 1766 cases, with a prevalence of 8.60%. In 2018, a total of 18,174 samples were included, including 1032 cases, with a prevalence of 5.68%.

As shown in Figure 1, the highest prevalence of diabetes was in 2015. The overall prevalence of the respondents was 8.60%, of which, the prevalence of male respondents was 7.44% and the prevalence of female respondents was 9.74%; the lowest prevalence of diabetes was in 2011, when the overall prevalence of the respondents was 5.30%, of which, the prevalence of male respondents was 4.68% and the prevalence of female respondents was 5.91%. In addition, the survey data showed that the prevalence of female respondents was higher than that of male respondents.

The survey respondents are stratified according to age groups, as shown in Figure 2, Figure 3, Figure 4 and Figure 5, which show that the age group with the lowest prevalence of respondents was 45 to 49 years old, the age groups with the highest prevalence of respondents were 60 to 64 years old and 65–69 years old, and the prevalence of female respondents was higher than that of male respondents in almost any age group.

The prevalence of diabetes in 2011, 2013, 2015, and 2018 were calculated according to the sampled 125 prefecture-level administrative regions, and visualized using ArcGIS 10.2. The results are shown in Figure 6.

In 2011, the prevalence of diabetes in the respondents was between 0.00% and 14.04%, and the prefecture-level cities with higher prevalence were mainly located in the northeast region and Beijing–Tianjin–Hebei region. In 2013, the prevalence of diabetes in the respondents was between 0.00% and 14.74%, and the prefecture-level cities with higher prevalence were mainly located in the central region, the northeast region and Beijing–Tianjin–Hebei region. In 2015, the prevalence of diabetes in the respondents was between 1.55% and 22.36%, and the prefecture-level cities with high prevalence were mainly located in the Beijing–Tianjin–Hebei region. In 2018, the prevalence of diabetes in the respondents was between 0.00% and 14.50%, and prefecture-level cities with high prevalence were distributed in the central region and the northeast region. The prevalence of diabetes is generally higher in the north than in the south, and in the coastal areas than in the inland [18].

### 3.2. Spatial Autocorrelation Analysis

Hotspot analysis was performed on the prevalence of diabetes of respondents in prefecture-level cities in 2011, 2013, 2015, and 2018, and their LISA maps were also made. The results are shown in Figure 7, Figure 8, Figure 9 and Figure 10, combined with global spatial autocorrelation (Table 1), showing that in 2011, 2013, 2015 and 2018, the prevalence of diabetes was clustered in China. The four-year prevalence hotspots appeared near the Beijing–Tianjin–Hebei region, and the Beijing–Tianjin–Hebei region has experienced high-value clusters of diabetes prevalence for four years according to LISA. However, Moran’s index decreased after 2013. Many hot and cold spots became not significant after 2013. High-High or Low-Low distribution areas also decreased slowly.

### 3.3. Analysis of Time and Space

Using SaTScan software to conduct a purely spatial analysis of the respondents in 2018 to accurately locate the spatial clustering area of diabetes, a Poisson distribution was used, and we set a maximum of 10% of the population in the at risk group. The results showed that the most likely clustering center appears in Cangzhou, Hebei Province. There were ten cities are in the dangerous areas (Cangzhou, Tianjin, Dezhou, Baoding, Binzhou, Beijing, Jinan, Shijiazhuang, Liaocheng, Weifang) (Table 2 and Figure 11), and 1899 respondents at risk.

In order to explore if diabetes had clustering characteristics in space and time, a spatiotemporal analysis of respondents in 2011, 2013, 2015, and 2018 was performed using SaTSca, with a maximum of 10% of the population at risk. The results showed that the most likely agglomeration center appears in Dezhou, Shandong Province. There are ten cities in the danger zone (Dezhou, Cangzhou, Jinan, Liaocheng, Binzhou, Shijiazhuang, Baoding, Tianjin, Puyang, Anyang) (Table 3 and Figure 12), and 1931 respondents at risk.

### 3.4. Binary Logistic Regression

In order to explore the factors that affect the occurrence of diabetes and assess the risk of diabetes, binary logistic regression was used for exploration based on the baseline data of 2018. The initial assignment of variables is shown in Table 4.

Table 5 shows the results of the chi-square test for single factors: age, location of residential address, education, hypertension, dyslipidemia, cancer, liver disease, smoking history, and alcohol use. A total of nine factors passed the chi-square test (*p* < 0.05) and could be included in binary logistic regression.

Binary logistic regression took diabetes as the dependent variable, age, location of residential address, education, hypertension, dyslipidemia, cancer, liver disease, kidney disease, smoking history, and alcohol use as independent variables. The Hosmer–Lemeshow test of the model was greater than 0.05 (0.889), indicating that the model had fully utilized the data and there was no very significant difference between the predicted value and the true value. Meanwhile, the result of the Omnibus test indicated that the model was statistically significant (*p* < 0.05). The established binary logistic regression can be expressed as Equation (5), according to Table 6.
(5)$$\begin{array}{ll} \ln\frac{p}{1-p}=-3.549 & -0.062*Age\left(50-54\right)+0.348*Age\left(55-59\right)+0.488\\ & *Age\left(60-64\right)+0.475*Age\left(65-69\right)+0.389*Age\left(\geq 70\right)\\ & +0.703*Hypertension+1.302*\mathrm{Dyslipidemia}+0.373\\ & *\mathrm{Kidney}\;\mathrm{Disease} \end{array}$$

The results showed that the occurrence of diabetes was significantly correlated with age, hypertension, dyslipidemia, kidney disease, and smoking history. The risk was higher in the 60–64 age group than in other age groups (OR = 1.635, *p* < 0.001). Patients with hypertension had a significantly higher risk of diabetes than those with other chronic diseases (OR = 2.004, *p* < 0.001). The highest risk was associated with dyslipidemia (OR = 3.598, *p* < 0.001).

### 3.5. Geographically Weighted Regression

Figure 13 showed the local R^2^ by using GWR (AICc = 640.402523, R^2^ = 0. 0.621877, Adjusted R^2^ = 0.609018). The distribution of residuals of GWR in space was randomized using Global Spatial Autocorrelation (*p* = 0.233661, spatial distribution model was random). Table 7 shows the statistics of local coefficient variables, illustrating that none of the factors exhibited significant spatial heterogeneity.

### 3.6. Disease Risk Assessment

Through binary logistic regression, we chose age, hypertension, dyslipidemia, cancer, heart attack, stroke, kidney disease, smoking history, and alcohol use as independent variables. We chose diabetes as the dependent variable to establish the binary logistic model and random forest model. AUC (area under the ROC curve) was used to evaluate the assessment model in this study. To verify whether the model’s expected risk result is consistent with the actual prevalence of diabetes, ArcGIS 10.4 was used to visualize the actual diabetes prevalence map and the diabetes risk assessment map (Figure 14), the high-risk assessment areas are mainly located in the Beijing–Tianjin–Hebei region and the northeast region. The random forest model’s assessment results are consistent with the actual prevalence, while the binary logistic regression model’s assessment results are far from the real incidence rate. Meanwhile, according to the ROC curve (Figure 15 and Figure 16), the accuracy of the random forest model (AUC = 0.7745) was higher than the binary logistic model (AUC = 0.6677). However, the random forest model cannot explain the function direction of independent variables and the relative risk degree of influencing factors, but binary logistic regression analysis can define the model and variables well.

## 4. Discussion

### 4.1. Innovation in This Study

Because the traditional data analysis method does not easily avoid interactions between the independent variables, as an emerging machine learning algorithm, the random forest algorithm performs well in avoiding multicollinearity. Therefore, it is widely used in the assessment of disease risk. The use of a random forest model to establish a concise and accurate diabetes risk assessment model is an innovative way to assess the risk of diabetes among people over 45 years old in China. Because the dataset does not always contain complete information, the distribution between positive and negative classes is mostly imbalanced, and some parameters are of low importance for the decision class, the random forest model performed better in this situation. We used the random forest model to make our diabetes risk assessment map, compared it with the assessment results of logistic regression, and noted that the assessment result was consistent with the actual prevalence. Thus, we conclude that the random forest model can achieve greater accuracy in assessing diabetes risk [41]. However, binary logistic regression analysis can intuitively explain diabetes risk factors, which is a disadvantage of the random forest model. The advantages of the two models should be combined in practical applications to allow them to jointly play a valuable role in disease risk assessment.

### 4.2. Scale Effect

The selection of different observation and analysis scales will result in the detection of different phenomena. This is known as the scale effect [42]. We took this into consideration when conducting our research. Our preliminary experiments showed that the spatial patterns obtained from the study at the prefecture-level city scale and the provincial scale are basically the same. Therefore, in order to get more detailed spatial patterns, our spatiotemporal analysis was based on the city-level prefecture scale.

### 4.3. Spatiotemporal Characteristic of Diabetes Prevalence

Diabetes prevalence remains high in China. According to the report from the International Diabetes Federation, diabetes prevalence in China had increased from 8.8% in 2011 to 10.9% in 2018 in adults 20–79 years. The prevalence of diabetes among people over 45 years old increased from 0.00% to 14.04% in 2011 to 0.00% to 14.50% in 2018 in the study area where the sample is located.

A significant Moran’s I test indicates that there is a presence of spatial autocorrelation, Getis and Ord’s G* could identify the hot or cold spot areas. Identifying hot spots for diseases is important for public health authorities who should adopt them for better-targeted interventions [43]. To determine the spatial patterns of a disease, local indicators of spatial association (LISA) in the environmental GIS are very helpful. This model is a set of methods used to describe and visualize spatial distributions, identify atypical locations or spatial outliers, determine patterns of spatial association, clusters, or hot-spots, and propose spatial regimes or other shapes of spatial heterogeneity [44].

In 2011, 2013, 2015, and 2018, the Moran’s I coefficient of diabetes prevalence in China was between 0.025585 and 0.104485, and showed non-random spatial distribution. Getis and Ord’s G* showed that hot spots are mostly found in the eastern and central regions, while cold spots are more common in southern regions. Local Spatial Autocorrelation analysis found that the High-High distribution pattern of diabetes is mainly found in cities close to the Beijing–Tianjin–Hebei region.

We also found that the spatial distribution model of diabetes was clustered, but that the tendency to cluster is waning, as the Moran’s I decreased from 0.103458 in 2011 to 0.025585 in 2018, and the hot and cold spot areas were also conspicuously decreased. Many areas also showed not significant High-High or Low-Low distributions.

The spatial scan statistic is a useful and widely used tool for detecting spatial or space–time clusters in disease surveillance. The software SaTScan, available for free, enhances this method’s ease-of-access for researchers [45]. We used SaTScan to accurately locate the spatial clustering areas of diabetes and to explore if diabetes had clustering characteristics in space and time.

Spatiotemporal clustering areas were detected by SaTScan software and they were located near the Beijing–Tianjin–Hebei region.

Therefore, diabetes prevalence has obvious spatial distribution characteristics in the population over 45 years old in China, that is, the north is higher than the south, the coast is higher than the inland, and economically developed areas are higher than economically underdeveloped areas. The specific reasons for the patterns need further research, but should be related to differences in eating habits and lifestyle changes caused by economic development, and by glycemic control, which varied greatly across geographic regions [46,47].

### 4.4. Diabetes Risk Factors

Binary logistic regression is often used to explore diabetes risk factors [48,49]. Binary logistic regression analysis showed that age, hypertension, dyslipidemia, and smoking history were all diabetes risk factors in this study.

In China, diabetes poses a severe threat to the population. Age is a main factor for diabetes [50]. In this study, especially after the age of 55, diabetes risk increased significantly with age. Therefore, middle-aged and elderly residents in China should always pay attention to their health, so as not to miss the best treatment time.

Besides, compared with other chronic diseases, hypertension and dyslipidemia are more likely to lead to diabetes, and diabetes also likely leads to the occurrence of hypertension or dyslipidemia [51,52,53]. As the main component of metabolic syndrome, diabetes, hyperglycemia, and hyperlipidemia interconnect and influence each other, forming a complex framework of chronic diseases [54]. With the prolongation of the disease’s course, the patient’s body’s immune function becomes increasingly abnormal, the function of many systems is weakened, and multiple diseases are prone to occur. With the prolongation of the disease’s course, the function of many systems in the patient’s body is weakened, which always leads to multiple diseases [55,56,57,58].

More and more studies show that smoking significantly increases the risk of diabetes [59]. Thus, diabetes patients with a history of smoking are reported to be at especially increased risk of incidence and poor outcomes from severe acute respiratory syndrome coronavirus [60]. China is one of the countries with the largest number of tobacco consumers in the world [61,62], which may be one of the reasons for the high prevalence of diabetes, and even of other chronic diseases, in China.

### 4.5. Spatial Heterogeneity of Diabetes Risk Factors

A GWR model is a simple and effective technology used to deal with spatial heterogeneity. Unlike traditional multiple linear regression, GWR lets regression parameters vary across space [63]. A GWR model was used to explore the spatial heterogeneity of diabetes risk factors. However, the results showed that there is no obvious spatial heterogeneity in the four risk factors (age, hypertension, dyslipidemia, and smoking history). This might be because this study did not incorporate socioeconomic and environmental factors into the study [64,65].

### 4.6. Limitations and Future Research

There are still some deficiencies in this research. For example, environmental factors, which are closely related to the prevalence of diabetes, have not been considered in this study. Besides, our approach to spatiotemporal analysis in this study was still traditional, and factors included in the model were not enough. In addition, there is still room for improvement in the accuracy of the model, and we are also trying to add other classification algorithms to our research. We will continue to advance this research, and it is believed that our research will provide accurate data support for improving the living conditions of people over 45 years old in China.

## 5. Conclusions

Firstly, in this paper, spatial autocorrelation and spatiotemporal clustering analysis were used to analyze the spatial distribution characteristics of diabetes. Secondly, we used the binary logistic regression model to explore the risk factors of diabetes in detail. Finally, the logistic regression model and random forest model were used to assess the risk of diabetes in people over 45 years old in China. The results showed that the clustering areas of patients with diabetes were mainly in the Beijing–Tianjin–Hebei region. The tendency to find clusters of diabetes prevalence among people over 45 years old in China is waning. Age, hypertension, dyslipidemia, and smoking history all had effects on diabetes, but the spatial heterogeneity of these factors were weak. Compared with the binary logistic model, the random forest model showed better fitness in assessing diabetes risk, and showed that the high-risk regions are the northeast region and the Beijing–Tianjin–Hebei region. Therefore, our method can analyze the spatial distribution characteristics and influencing factors of diabetes, but there is still room for improvement in the accuracy of assessing the risk of diabetes. We will continue to follow up on this study after the data of CHARLS is updated, and we will explore more excellent methods in the following research.

## Figures and Tables

**Figure 1 ijerph-19-09861-f001:**
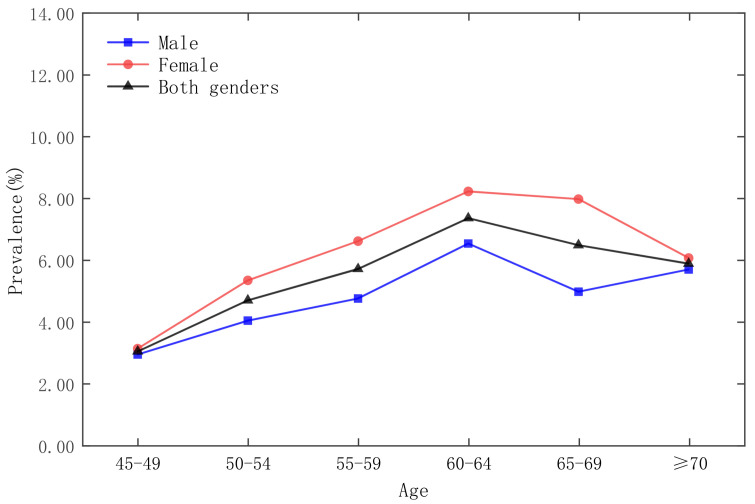
Prevalence of diabetes by gender in 5-year age groups in the CHARLS 2011 national survey.

**Figure 2 ijerph-19-09861-f002:**
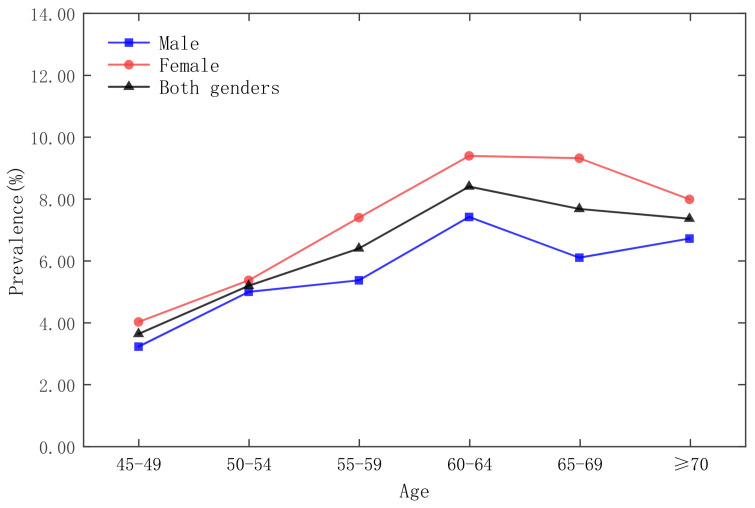
Prevalence of diabetes by gender in five-year age groups in the CHARLS 2013 national survey.

**Figure 3 ijerph-19-09861-f003:**
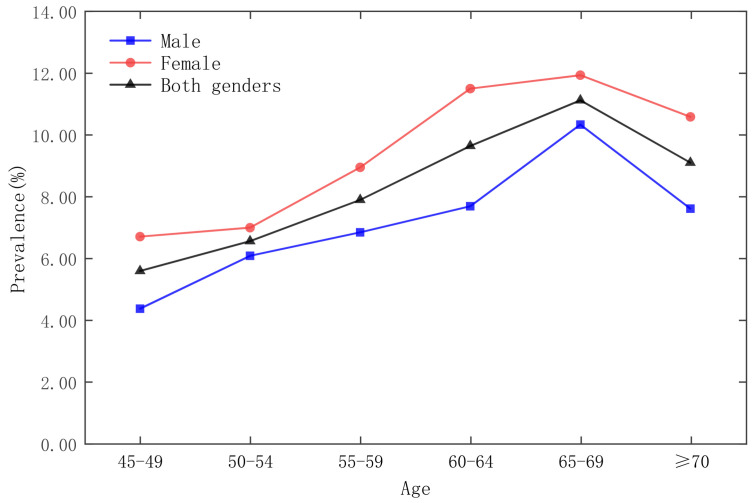
Prevalence of diabetes by gender in five-year age groups in the CHARLS 2015 national survey.

**Figure 4 ijerph-19-09861-f004:**
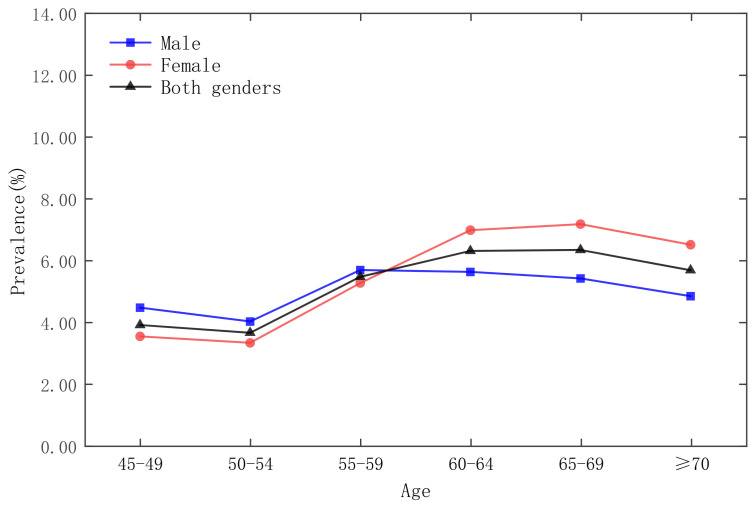
Prevalence of diabetes by gender in five-year age groups in the CHARLS 2015 national survey.

**Figure 5 ijerph-19-09861-f005:**
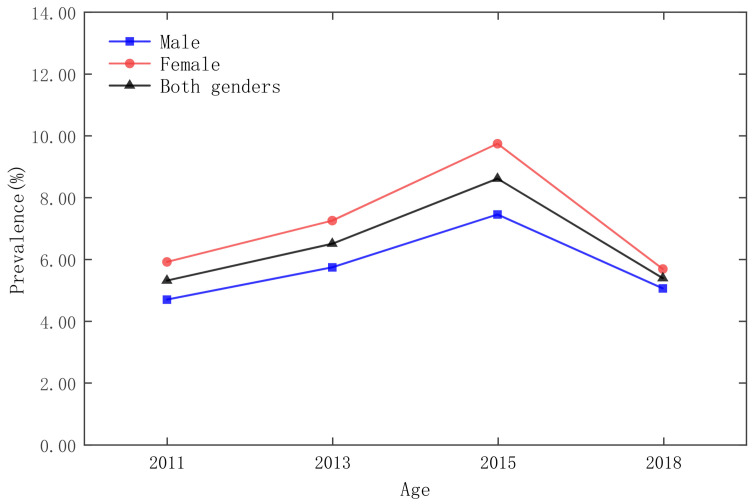
Prevalence of diabetes by age in 2011, 2013, 2015, and 2018.

**Figure 6 ijerph-19-09861-f006:**
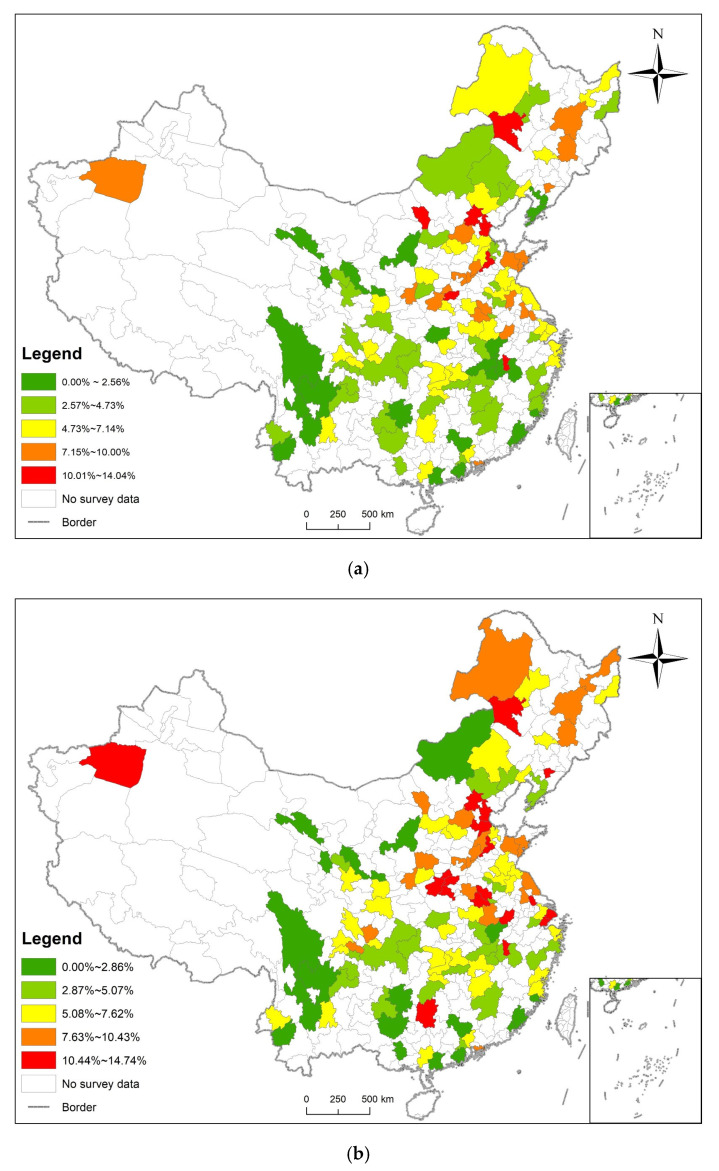
Prevalence of diabetes visualized. (**a**) The prevalence of diabetes in 2011 was divided into five classifications according to the natural breaks method; (**b**) The prevalence of diabetes in 2013 was divided into five classifications according to the natural breaks method; (**c**) The prevalence of diabetes in 2015 was divided into five classifications according to the natural breaks method; (**d**) The prevalence of diabetes in 2018 was divided into five classifications according to the natural breaks method.

**Figure 7 ijerph-19-09861-f007:**
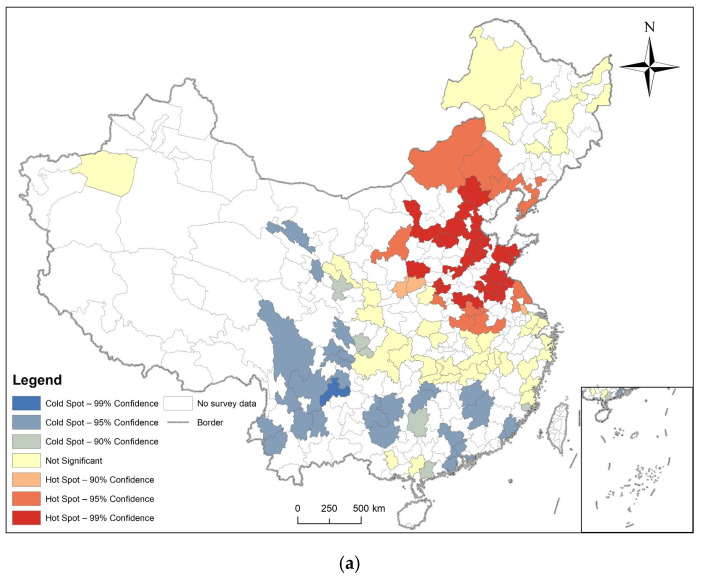
Spatial autocorrelation analysis of diabetes prevalence in 2011. (**a**) The result of Getis and Ord’s G*; (**b**) The result of local spatial autocorrelation analysis.

**Figure 8 ijerph-19-09861-f008:**
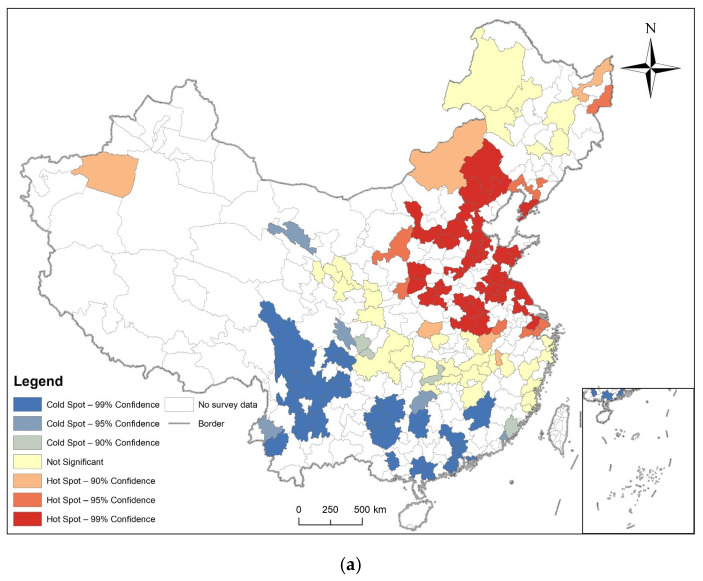
Spatial autocorrelation analysis of diabetes prevalence in 2013. (**a**) The result of Getis and Ord’s G*; (**b**) The result of local spatial autocorrelation analysis.

**Figure 9 ijerph-19-09861-f009:**
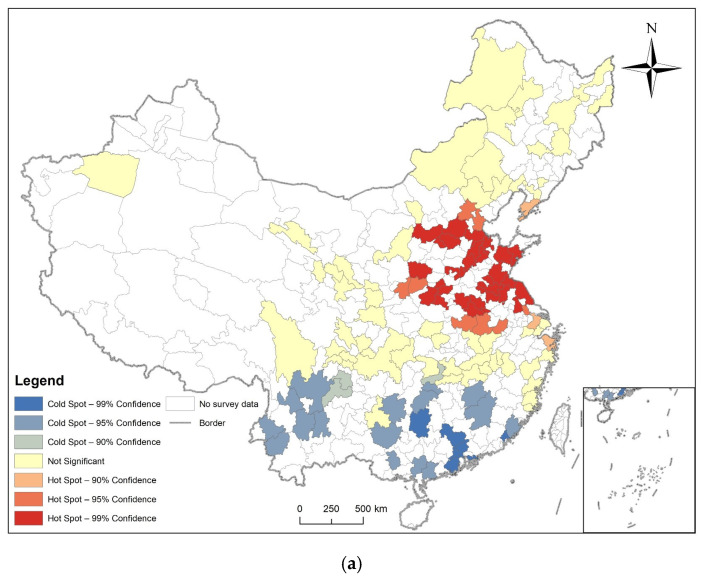
Spatial autocorrelation analysis of diabetes prevalence in 2015. (**a**) The result of Getis and Ord’s G*; (**b**) The result of local spatial autocorrelation analysis.

**Figure 10 ijerph-19-09861-f010:**
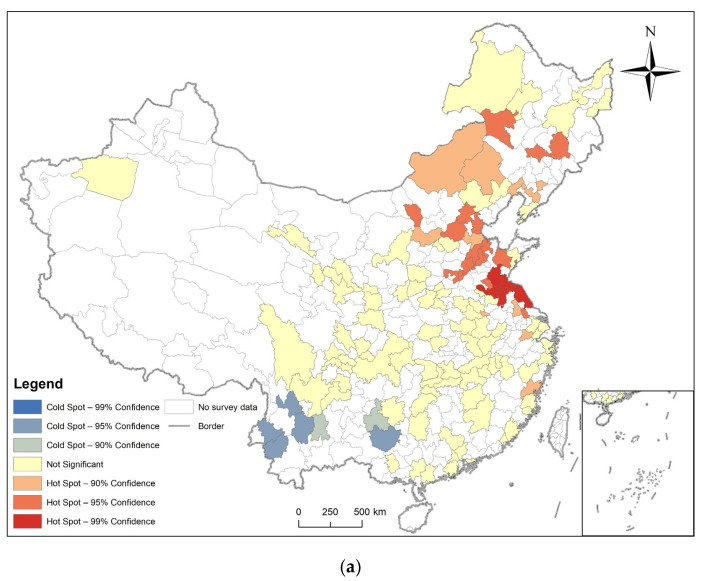
Spatial autocorrelation analysis of diabetes prevalence in 2018. (**a**) The result of Getis and Ord’s G*; (**b**) The result of local spatial autocorrelation analysis.

**Figure 11 ijerph-19-09861-f011:**
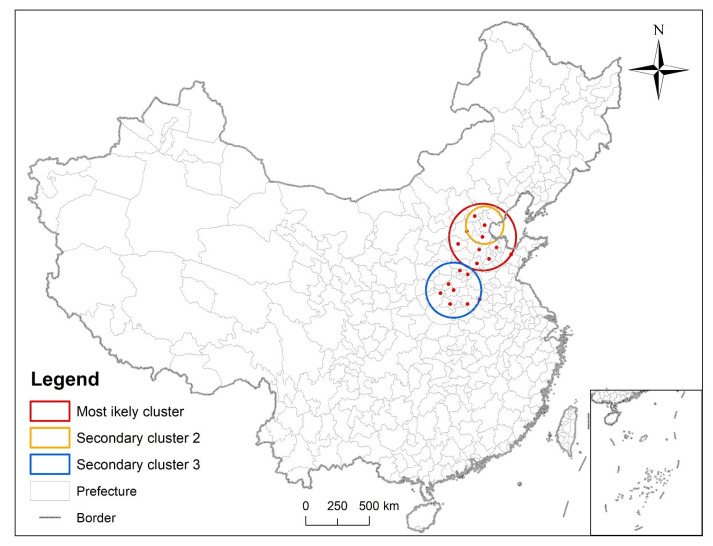
Three clusters were detected by purely spatial analysis.

**Figure 12 ijerph-19-09861-f012:**
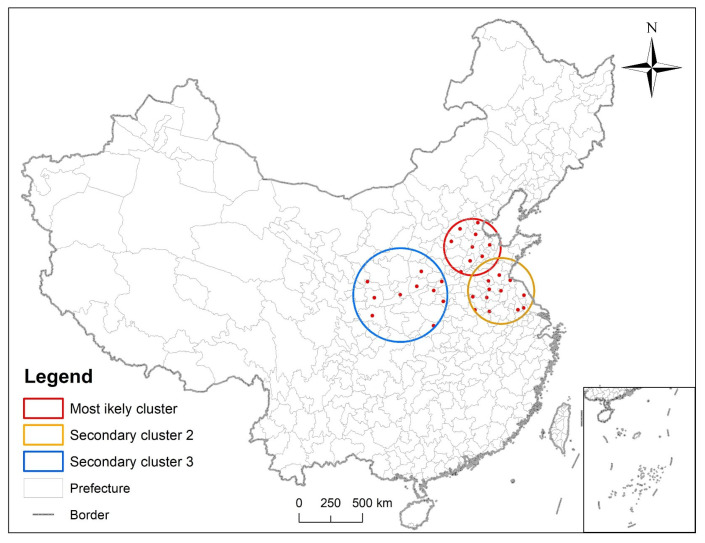
Three clusters were detected by spatiotemporal analysis.

**Figure 13 ijerph-19-09861-f013:**
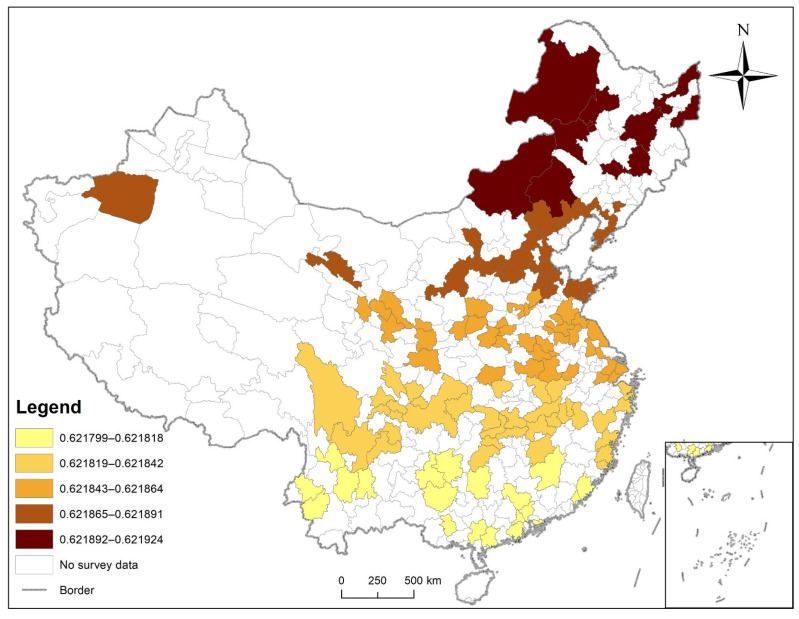
The local R^2^ of GWR.

**Figure 14 ijerph-19-09861-f014:**
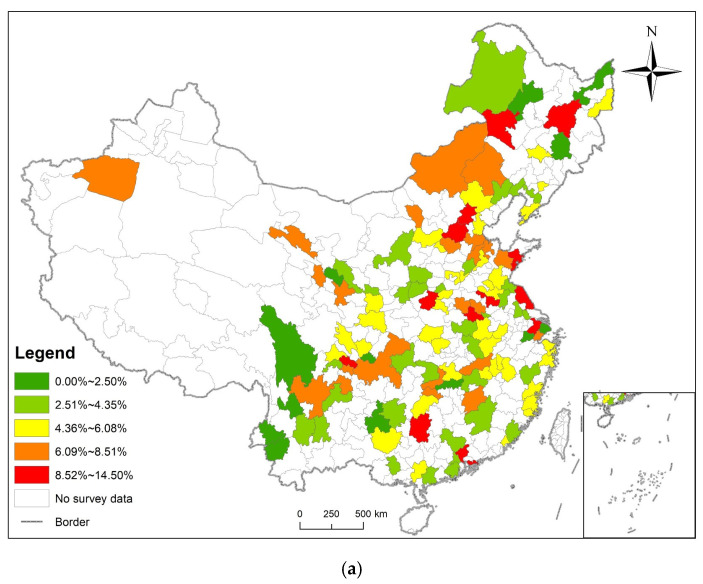
Assessment result. (**a**) The prevalence of diabetes in 2018 was divided into five classifications according to the natural breaks method. (**b**) The disease risk assessment result of binary logistic regression model was divided into five classifications according to the natural breaks method; (**c**) The disease risk assessment result of random forest model was divided into five classifications according to the natural breaks method.

**Figure 15 ijerph-19-09861-f015:**
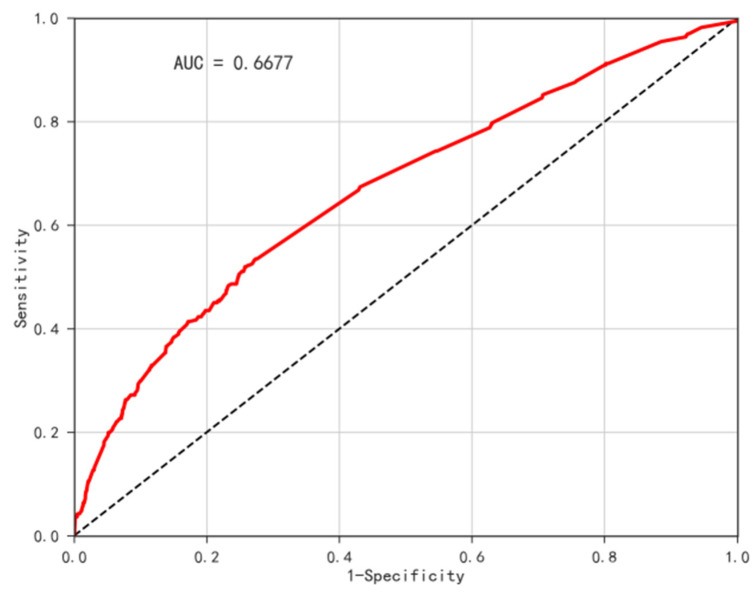
ROC Curve of binary logistic regression model.

**Figure 16 ijerph-19-09861-f016:**
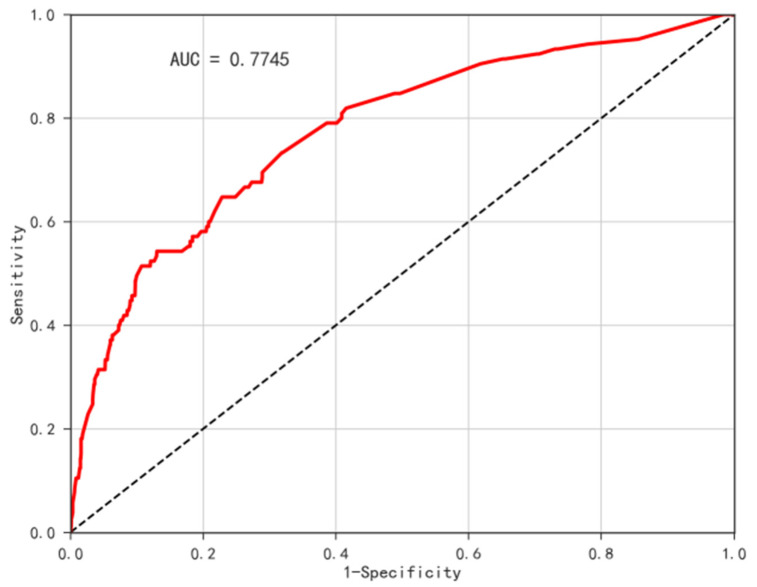
ROC Curve of random forest model.

**Table 1 ijerph-19-09861-t001:** Global spatial autocorrelation.

Date	Moran’s Index	*p*-Value	Z-Score	Spatial Distribution Model
2011	0.103458	<0.001	7.139808	Cluster
2013	0.104485	<0.001	7.205062	Cluster
2015	0.067174	<0.001	4.835403	Cluster
2018	0.025585	<0.007	2.652944	Cluster

**Table 2 ijerph-19-09861-t002:** Purely spatial analysis results by using SaTScan software.

Cluster Center	Radius (km)	Region	Logarithmic Likelihood Ratio	Relative Risk Level	*p*-Value
Cangzhou, Hebei Province	270.98	Cangzhou, Tianjin, Dezhou, Baoding, Binzhou, Beijing, Jinan, Shijiazhuang, Liaocheng, Weifang	52.819422	1.54	<0.001
Tianjin	153.02	Tianjin, Cangzhou, Beijing, Baoding	41.161335	1.78	<0.001
Zhengzhou, Henan Province	221.64	Zhengzhou, Jiaozuo, Luoyang, Pingdingshan, Zhoukou, Anyang, Puyang, Bozhou	39.852687	1.54	<0.001

**Table 3 ijerph-19-09861-t003:** Spatiotemporal analysis results by using SaTScan software.

Cluster Center	Radius (km)	Region	Logarithmic Likelihood Ratio	Relative Risk Level	*p*-Value
Dezhou, Shandong Province	229.44	Dezhou, Cangzhou, Jinan, Liaocheng, Binzhou, Shijiazhuang, Baoding, Tianjin, Puyang, Anyang	163.632756	4.16	<0.001
Suqian, Jiangsu Province	264.81	Suqian, Xuzhou, Lianyungang, Suzhou, Linyi, Zaozhuang, Yancheng, Huainan, Yangzhou, Taizhou, Bozhou, Fuyang, Hefei	109.037860	3.39	<0.001
Weinan, Shanxi Province	377.23	Weinan, Yuncheng, Baoji, Linfen, Luoyang, Hanzhong, Pingliang, Pingdingshan, Jiaozuo, Xiangfan, Zhengzhou	94.209061	3.20	<0.001

**Table 4 ijerph-19-09861-t004:** Variables and assignments.

Variables	Type	Assignments
Gender	Integer	0 = Male; 1 = Female
Age	Integer	0 = 45–49; 1 = 50–54; 2 = 55–59; 3 = 60–64; 4 = 65–69; 5 = 70 or more
Location of Residential Address	Integer	0 = Central of City/Town; 1 = Urban-Rural Integration Zone; 2 = Rural; 3 = Special Zone
Education	Integer	0 = Illiterate; 1 = Did not Finish Primary School; 2 = Sishu/Home School; 3 = Elementary School; 4 = Middle School; 5 = High School; 6 = Vocational School; 7 = Two-/Three-Year College/Associate Degree; 8 = Four-Year College/Bachelor’s Degree or more
Marital Status	Integer	0 = Married with Spouse Present; 1 = Married but Not Living with Spouse Temporarily for Reasons Such as Work; 2 = Separated; 3 = Divorced; 4 = Widowed; 5 = Never Married
Nation	Integer	0 = Han Nationality; 1 = Zhuang Nationality; 2 = Manchu; 3 = Hui Nationality; 4 = Miao Nationality; 5 = Uyghur Nationality; 6 = Tujia Nationality; 7 = Yi Nationality; 8 = Other Nationality
Hypertension	Integer	0 = No; 1 = Yes
Dyslipidemia	Integer	0 = No; 1 = Yes
Diabetes	Integer	0 = No; 1 = Yes
Cancer	Integer	0 = No; 1 = Yes
Liver Disease	Integer	0 = No; 1 = Yes
Emotional Problems	Integer	0 = No; 1 = Yes
Smoking History	Integer	0 = No; 1 = Yes
Alcohol Use	Integer	0 = No; 1 = Yes

**Table 5 ijerph-19-09861-t005:** Chi-square test result.

Factors		The Total Number of Samples	Number of Cases		X2	*p*-Value
Gender					3.734	0.053
	Male	8715	463	5.31%		
	Female	9459	569	6.02%		
Age					37.133	<0.001
	45–49	1307	53	4.06%		
	50–54	3173	120	3.78%		
	55–59	3135	181	5.77%		
	60–64	2856	192	6.72%		
	65–69	3063	207	6.76%		
	≥70	4640	279	6.01%		
Location of Residential Address					13.003	0.005
	Central of City/Town	3486	232	6.66%		
	Urban-Rural Integration Zone	1270	90	7.09%		
	Rural	13346	706	5.29%		
	Special Zone	72	4	5.56%		
Education					17.018	0.03
	Illiterate	4022	252	6.27%		
	Did not Finish Primary School	3764	208	5.53%		
	Sishu/Home School	41	2	4.88%		
	Elementary School	4030	221	5.48%		
	Middle School	4023	201	5.00%		
	High School	1503	81	5.39%		
	Vocational School	420	35	8.33%		
	Two-/Three-Year College/ Associate Degree	229	22	9.61%		
	Four-Year College/Bachelor’s Degree or more	142	10	7.04%		
Marital Status					3.355	0.645
	Married with Spouse Present	14281	820	5.74%		
	Married But Not Living with Spouse Temporarily for Reasons Such as Work	1214	63	5.19%		
	Separated	65	4	6.15%		
	Divorced	226	9	3.98%		
	Widowed	2280	133	5.83%		
	Never Married	108	3	2.78%		
Nation					10.489	0.232
	Han Nationality	17077	975	5.71%		
	Zhuang Nationality	177	8	4.52%		
	Manchu	301	12	3.99%		
	Hui Nationality	107	12	11.21%		
	Miao Nationality	112	3	2.68%		
	Uyghur Nationality	81	6	7.41%		
	Tujia Nationality	25	1	4.00%		
	Yi Nationality	97	3	3.09%		
	Other Nationality	197	12	6.09%		
Hypertension					161.428	<0.001
	No	16273	792	4.87%		
	Yes	1901	240	12.62%		
Dyslipidemia					433.646	<0.001
	No	16601	739	4.45%		
	Yes	1573	293	18.63%		
Cancer					8.651	0.003
	No	17946	1008	5.62%		
	Yes	228	24	10.53%		
Liver Disease					12.350	<0.001
	No	17603	979	5.56%		
	Yes	571	53	9.28%		
Emotional Problems					2.246	0.134
	No	17968	1015	5.65%		
	Yes	206	17	8.25%		
Smoking History					19.540	<0.001
	No	17359	955	5.50%		
	Yes	815	77	9.45%		
Alcohol Use					11.566	0.001
	No	11936	731	6.12%		
	Yes	6238	301	4.83%		

**Table 6 ijerph-19-09861-t006:** Binary logistic regression analysis.

Variables	B	SE	Wald	Df	*p*-Value	OR	95% CI	
							Lower	Upper
Age (45–49)			31.808	5	0.000			
50–54	−0.062	0.170	0.134	1	0.714	0.939	0.673	1.311
55–59	0.348	0.162	4.629	1	0.031	1.416	1.031	1.944
60–64	0.488	0.161	9.193	1	0.002	1.629	1.188	2.232
65–69	0.475	0.160	8.843	1	0.003	1.607	1.176	2.198
≥70	0.389	0.155	6.291	1	0.012	1.475	1.089	2.000
Hypertension	0.703	0.081	75.339	1	0.000	2.020	1.723	2.367
Dyslipidemia	1.302	0.076	295.059	1	0.000	3.676	3.169	4.265
Smoking history	0.373	0.128	8.446	1	0.004	1.452	1.129	1.867
Constant	−3.549	0.144	606.838	1	0.000	0.029		

**Table 7 ijerph-19-09861-t007:** Coefficient of risk factors.

Variables	Mean	Max	Min
Age	0.0571	0.057091	0.05713
Hypertension	0.007537	0.007361	0.00765
Dyslipidemia	0.265775	0.26573	0.265843
Smoking History	0.00879	0.008752	0.008823

## Data Availability

“China Health and Retirement Longitudinal Study” at http://charls.pku.edu.cn/ (accessed on 28 May 2022).

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
