# Peer review of "Spatiotemporal Analysis and Risk Assessment Model Research of Diabetes among People over 45 Years Old in China"

_ijerph, 2022, doi:10.3390/ijerph19169861_

Round 1
Reviewer 1 Report
Reviewer comments and suggestions
In this study, we used spatial autocorrelation, spatiotemporal cluster analysis, binary logistic regression, and random forest model, to analyze the spatiotemporal distribution and possible risk factors for diabetes among people over 45 years old in Chinese and try to assess diabetes risk
The result showed while using binary logistic regression analysis age, hypertension dyslipidemia cancer heart attack, stroke, kidney disease, ever smoked and alcohol were all diabetes risk factors (P < 0.05). The authors also observed that the random forest model showed a better fitness in assessing diabetes risk, that the fitting accuracy is 77.45% of the random forest model. Based on this result model, the Northeast region and the Beijing-Tianjin-Hebei region are high-risk areas for diabetes. These results provide benchmark data for the prevention and control of diabetes.
For the improvement of the manuscript, I am suggesting a few comments.
- Line 39 need to mention more disorder with references
- Line 62-63 please cite a few manuscripts, line 71 what does it mean whole china
- Line 82-83 The authors need to provide other information as well
- Line 97 Please define Moran index for the common reader of your paper
- Table 5 It would be nice if the authors describe the participants they recruited in every year
- Table 6 The authors could use an adjusted model here
- General sentences, it would be better to present the novelty of this study in the first para of the discussion
- Line 299-302 I do not think the statement could be important as 45 year old is not an aging population.
- Line 310 I am in doubt about from 0 % to 14%. Did the author study about incidence cases. please check the statement and correct it
- Line 330-334 please avoid big sentences
- Line 340 ending line does not seem to fit here
- Line 343-344 What do the authors want to say here
- Line 355-356 please cite the reference and line 358 reference ( needed)
- Line 359-360 please valid the points with other studies and the reason for this
- All references need to be modified based on the MDPI journals.
Reviewer 2 Report
1. Even though the data come from the survey, the representativeness of the sample should be elaborated upon to enhance the conclusion.
2. The methodology is not sufficiently novel; why not utilize the MLM or GWR model to investigate contextual effects or spatial heterogeneity?
3. The assessment of the work is quite weak; it could be reinforced by a description and discussion of the most recent advancements and limitations in the field.
4. The analysis should be improved for various calculation outcomes, such as examining the spatial and temporal characteristics of diabetes incidence, the local anomalous distribution, the random distribution in 2018, etc.
5. The paper's discussion section could be strengthened, and the spatial heterogeneity of diabetes incidence, advantages and disadvantages of research methodologies, scale effect, etc. should be explored in detail.
6. The research scale is city-level prefecture. Does the conclusion exist MAUP effect?
7. The 2018 data was utilized in both Figure 11 and Figure 12. What is the distinction?
8. The style of the abstract should be altered to emphasize the study's findings and crucial data indications.
9. sSpatial → Spatial.
Round 2
Reviewer 1 Report
No more comments. All comments were addressed. However, still I find mistake in the references. Please check it again
Reviewer 2 Report
1. The discussion section is still not clear, the author should edit the text according the writing style in published paper for better understanding. For exmaple, what's the meaning in L338-343 for the discussion?
2. The author should add the MLM or GWR model to investigate contextual effects or spatial heterogeneity as the future works.
Author Response
Please see the attachment.

This manuscript is a resubmission of an earlier submission. The following is a list of the peer review reports and author responses from that submission.